# Frankenstein, thematic analysis and generative artificial intelligence: Quality appraisal methods and considerations for qualitative research

Tanisha Jowsey [1]*, Peta Stapleton[2], Shawna Campbell[2], Alexandra Davidson [3], Cher McGillivray[2], Isabella Maugeri [1], Megan Lee[2], Justin Keogh [1]

**1** Faculty of Health Sciences and Medicine, Bond University, Gold Coast, Australia, **2** Faculty of Society and Design, Bond University, Gold Coast, Australia, **3** Institute for Evidence-Based Healthcare, Bond University, Gold Coast, Australia

\* tjowsey@bond.edu.au

## Abstract

### Objective

To determine accuracy and efficiency of using generative artificial intelligence (GenAI) to undertake thematic analysis.

### Introduction

With the increasing use of GenAI in data analysis, testing the reliability and suitability of using GenAI to conduct qualitative data analysis is needed. We propose a method for researchers to assess reliability of GenAI outputs using deidentified qualitative datasets.

### Methods

We searched three databases (United Kingdom Data Service, Figshare, and Google Scholar) and five journals (PlosOne, Social Science and Medicine, Qualitative Inquiry, Qualitative Research, Sociology Health Review) to identify studies on health-related topics, published prior to whereby: humans undertook thematic analysis and published both their analysis in a peer-reviewed journal and the associated dataset. We prompted a closed system GenAI (Microsoft Copilot) to undertake thematic analysis of these datasets and analysed the GenAI outputs in comparison with human outputs. Measures include time (GenAI only), accuracy, overlap with human analysis, and reliability of selected data and quotes.

### Results

Five studies were identified that met our inclusion criteria. The themes identified by human researchers and Copilot showed minimal overlap, with human researchers often using discursive thematic analyses (40%) and Copilot focusing on thematic

which permits unrestricted use, distribution, and reproduction in any medium, provided the original author and source are credited.

**Data availability statement:** Some data are held in the UK Data Service repository, four of the five datasets have safeguarded restrictions. Dataset 1 (Barlow et al.) ISSN 0277-9536, https://doi.org/10.1016/j.socscimed.2021.113761. Dataset 2 (Hervey & Antova) UK Data Service SN: 854778, DOI: 10.5255/UKDA-SN-854778 Dataset 3 (Dunn et al) UK Data Service SN: 854245, DOI: 10.5255/UKDA-SN-854245 Dataset 4 (Holman & Walker) UK Data Service SN: 855082, DOI: 10.5255/UKDA-SN-855082 Dataset 5 (Arora) UK Data Service. SN: 855953, DOI: 10.5255/UKDA-SN-855953.

**Funding:** The author(s) received no specific funding for this work.

**Competing interests:** The authors have declared that no competing interests exist.

analysis (100%). Copilot's outputs often included fabricated quotes (58% SD = 45%) and none of the Copilot outputs provided participant spread by theme. Additionally, Copilot's outputs primarily drew themes and quotes from the first 2-3 pages of textual data, rather than from the entire dataset. Human researchers provided broader representation and accurate quotes (79% quotes were correct, SD = 27%).

## Conclusions

Based on these results, we cannot recommend the current version of Copilot for undertaking thematic analyses. This study raises concerns about the validity of both human-generated and GenAI-generated qualitative data analysis and reporting.

## Introduction

*"So much has been done, exclaimed the soul of Frankenstein – more, far more, will I achieve; treading in the steps already marked, I will pioneer a new way, explore unknown powers, and unfold to the world the deepest mysteries of creation."* – Mary Shelley [1].

Mary Shelley's *Frankenstein* (1831) is a classical text warning society of the potential for conflict that stems from human failure to recognize that actions have repercussions (and that humans are not gods). It is also a story about a monster created by a well-meaning scientist [1]. In this article, we conjure up such realities in an analysis of the potential – and associated repercussions – of Generative Artificial Intelligence (GenAI) large language models (LLMs) for qualitative analysis. Presently, GenAI can be used to undertake social science analysis according to seemingly any epistemology or interpretive framework. This creates the possibility of a GenAI-augmented analysis, which several researchers have recently explored with Chat-GPT [2–7]. Using GenAI to undertake qualitative analysis holds the potential to exponentially speed up analysis from years to seconds [8].

Tensions exist. Kidder and Fine (1987) describe a distinction between 'small q qualitative' and 'Big Q Qualitative' scholarship [9]. Small q qualitative scholarship is proceduralist or technique-focused qualitative research that is often applied within positivist, mixed-method, and deductive research projects. Examples of small q qualitative research abound within medical and health sciences domains. Whereas Big Q qualitative research is concerned with alignment between researcher values and epistemologies to the methods they undertake and the ways in which findings are presented. Big Q qualitative research is open to creative methodological approaches such as discursive or "epistemologically radical" forms [10] and is rarely published in medical and health sciences domains. We argue that many Big Q qualitative approaches – such as grounded theory and phenomenology studies – cannot or should not be delegated to GenAI. Even multi-modal a/r/t-o-graphical or autoethnographic approaches would need very careful consideration of how to engage with GenAI in productive and rigorous ways.

Fiery debate between social scientists concerns whether quality reporting checklists such as the Consolidated Criteria for Reporting Qualitative Research (COREQ) and Standards for Reporting Qualitative Research (SRQR) are appropriate and support methodological congruence. Several leading social scientists have warned that checklist and reporting requirements of journals (such as use of the COREQ and SRQR or Introduction, Methods, Results and Discussion (IMRaD) format of abstracts) create rigidity that (somewhat ironically) inhibits transparency [10], and can cause what Sparkes and Smith have called *methodolatry* [11].

Could GenAI be a new source of methodolatry? Or is it the godsend that small q qualitative researchers have been waiting for? Given the popularity – and even requirements – of COREQ, SRQR and other checklist use in medical and health science journals, small q qualitative researchers should be asking: what are the risks and opportunities for qualitative research that GenAI poses in terms of quality, rigour, transparency, and trustworthiness? In this paper, we consider trustworthiness specifically in relation to thematic analysis, which is a popular qualitative approach in medical and health sciences [12]. Thematic analysis has been described by Braun and Clarke, and others as a method used to identify and report recurring thematic patterns in qualitative data [13–16].

Prior to the release of Chat-GPT 3.5, there was a rich body of social science literature – both small q and Big Q – exploring analytical trustworthiness [17–20]. Trustworthiness is achieved through consistency and transparency about the methods undertaken including analytical processes, and involves such processes as data reduction (selecting, concentrating, simplifying, structuring, and converting data) [21] and reflexivity [19]. When multiple social scientists analyze the same dataset, they might identify different points of significance [18]. Perfect replicability is not the goal in qualitative approaches, as researchers acknowledge the role of subjectivity and context. Instead, the focus of small q qualitative research is increasingly on demonstrating consistency, transparency, and trustworthiness in the analytical process [10,13,22–25]. The specific combination of reliability techniques used may vary depending on the research paradigm and methodological approach adopted. In medical and health science scholarship, qualitative researchers often work in teams, undertaking member checking [22] and group discussions to agree on the themes most relevant to the research question and most reflective of the data.

When Chat-GPT 3.5 was released and scientists started exploring its functionality for analysis, the issue of transparency quickly came to the foreground. As with many GenAI software, Chat-GPT 3.5 outputs were not accompanied by transparent evidence of how GenAI arrived at the outputs. Many GenAI software developers, such as Avidnote have since worked on ways to increase transparency, which is reassuring for social scientists. These are, however, usually behind a paywall. To date, few studies have compared human research outputs with GenAI outputs on existing qualitative datasets. Morgan compared Chat-GPT with human researchers on two existing studies and found Chat-GPT performed "reasonably well" at identifying descriptive themes from the two datasets. Morgan suggested GenAI could reduce the burden on human researchers for manual comparison coding of qualitative data [26]. Echoing this, Bennis & Mouwafaq used nine GenAI models (Llama 3.1 405B, Claude 3.5 Sonnet, NotebookLM, Gemini 1.5 Advanced Ultra, ChatGPT o1-Pro, ChatGPT o1, GrokV2, DeepSeekV3, Gemini 2.0 Advanced), and found that overall GenAI tools are helpful to reduce time to analyse qualitative data [27]. However, little is known in the literature about other tools, including Microsoft's Copilot. This is despite Copilot's superiority in data governance and security due to its 'closed' nature [28], potentially making it a safer choice for analyzing sensitive human data.

We wanted to know if evidence supports the use of general purpose LLMs for undertaking thematic analysis, and if so, how such use contributes to Big Q and little q discourse. We aimed to build on this small body of evidence from Morgan and Bennis & Mouwafaq to determine accuracy and efficiency of using a general-purpose LLM to undertake thematic analysis [26,27]. We asked: To what extent does Copilot provide reliable thematic analysis of existing previously published qualitative datasets?

To evaluate the reliability of thematic analysis outputs generated by GenAI, we identified thematic analysis studies whereby human researchers had published a thematic analysis and published the associated dataset. We prompted GenAI to analyze these same datasets and then compared human and GenAI outputs.

## Materials and methods

The GenAI tool we selected was Copilot on the grounds that it was a general-purpose GenAI, approved for use by both our institution and the UK Data Service, and was a closed GenAI system (i.e., access to Copilot is limited and password protected, so that the data are protected and are not used to train foundation GenAI models). The UK Data Service permitted us to include in our study data published through their repository but only for analysis through approved closed GenAI systems (i.e., Copilot) and not for open GenAI systems like ChatGPT, Perplexity, Claude or Gemini. We conducted comparative analysis of the themes and supporting evidence provided by human researchers in their published paper versus GenAI. We appraised outputs in terms of transparency, and trustworthiness, and also with reference to the COREQ checklist. Our comparative analyses were all conducted by human researchers without the aid of GenAI in any of our processes.

## Search strategy

In August and September 2024, we conducted a comprehensive search across three databases—UK Data Service, Figshare, and Google Scholar—and five relevant academic journals, namely PLOS ONE, Social Science & Medicine, Qualitative Inquiry, Qualitative Research, and Sociology of Health Review. Our objective was to identify studies published prior to 2022 (before the boom in GenAI precipitated by the release of freely available Open AI Chat-GPT), in which researchers performed thematic analysis, subsequently publishing their datasets, and publishing their associated analyses in peer-reviewed journals. Eligibility criteria assessing the participants, concept and context of published studies are presented in Table 1. We then prompted Copilot to perform thematic analyses on the identified datasets and subsequently analysed the outputs generated by the GenAI. The metrics assessed included time taken for analysis, accuracy of the results, overlap with human-generated analyses, and the reliability of selected data and quotations.

Studies were included if the analysis was published prior to 2022, available in English, and included thematic analysis. A subset of thematic analysis is discursive thematic analysis, whereby analysts consider how socio-cultural contexts inform the language (i.e., discourse) presented through textual data (i.e., spoken or written texts). Only peer reviewed published studies were included. The following also applied:

- Exclude studies not concerning healthcare

- Exclude studies that do not apply a thematic analysis

- Exclude animal studies

- Exclude simulation studies and education studies conducted outside of the healthcare or health policy setting

**Table 1. PICO.**

|  | Inclusion |
| --- | --- |
| Population | Studies concerned with adult or paediatric humans: patient, informal carer, family and/or healthcare provider. |
| Intervention | Study topic concerned with healthcare and health policy |
| Context | Healthcare or health policy |
| Outcome | Stand-alone qualitative research or a mixed-methods study including thematic analysis |
| Other | Findings are reported in an international peer-reviewed journal and associated qualitative dataset is published. Study published before 2022 (to reduce likelihood that study authors have used GenAI to undertake original analysis) Geographic Location: Open Cultural/sub-cultural factors: all included Specific racial or gender-based interests: all included Full-length journal articles Database: UK Data Service or Figshare |

- Exclude protocols, editorials, commentaries and opinion papers

- Exclude unpublished studies/ grey literature

- Exclude reviews

## Study selection

Studies were identified according to inclusion criteria. Two authors (TJ & JK) searched for studies in UK Data Service and Figshare, and in these journals: Social Science and Medicine, Qualitative Inquiry, Qualitative Research, and Sociology Health Review. Search terms were 'thematic analysis AND healthcare' filtered by year 2000–2021. Where studies were identified via UK Data Service or Figshare, we then searched on Google Scholar, using key author names (authors in position 1, 2, or final author) and key word from the dataset title to identify published thematic analysis studies that reported on the identified dataset. Where studies were identified via journals we searched for hyperlinked supplementary files of qualitative datasets. We identified four studies through the UK Data Service that met our inclusion criteria, so sought and obtained UK Data Service permission to analyze Copilot capability using these studies. We also sought UK Data Service permission to use Perplexity (an open GenAI system) to compare with Copilot, however this was rejected due to data security concerns. One of the four studies identified through the UK Data Service [31] was included, although the authors only made 15 of 54 interviews available through the repository. We uploaded these 15 interviews in Copilot for analysis.

Our prompt did not include a specific discursive component

## Data extraction

We worked online as a group at the same time to enter incognito onto Copilot and uploaded each dataset along with a structured zero-shot prompt (Table 2). A zero-shot prompt is one that provides no examples and relies on the LLM's pretrained knowledge to interpret and complete the task. Prior to running the prompt through Copilot, the team developed, tested, refined, reviewed, and piloted the prompt wording. We ran the prompt three times for each dataset. Each time we completed the prompt-output and saved the output; we closed down Copilot completely and then reopened it to run the next prompt. We recorded time taken to produce each output and then appraised the first output for each study. Each output was appraised by two independent researchers. We then met as a whole team to discuss methods and results.

We appraised both Copilot outputs and human researcher published articles; we identified markers of rigour and trustworthiness, extracted themes, and checked every reported quote against its related published dataset. When checking quotes for accuracy against source material, we searched for each output quote in total and in parts (three consecutive words and key word searches). In instances where we could not identify any partial sentences or similar sentences to the reported quote we deemed this fabricated.

For markers of rigour and transparency we drew on the Consolidated criteria for reporting qualitative research framework (COREQ) [29]. COREQ assesses the rigour of qualitative research, such as characteristics of the research team,

**Table 2. GenAI Prompt.**

*Undertake thematic analysis of the dataset I have uploaded alongside this prompt. Review the dataset in its provided format. Begin by familiarizing yourself with the content and noting key recurring ideas. Generate initial codes based on identified features and apply these codes consistently across the dataset. Group related codes into potential themes, using pattern detection algorithms if applicable. Evaluate and refine the themes to ensure they are coherent and relevant. Develop detailed descriptions and clear names for each theme. Prepare a comprehensive report summarizing the thematic findings, supported by relevant data excerpts. Ensure consistency in coding and theme development, validate the accuracy of themes, and document the entire process transparently. Provide between four and eight themes, supporting the themes with the number of participants that are represented in each theme. Provide quotes from participants to support each theme. Provide this thematic analysis summary within 800 words.*

reflexivity practices used in the research, the strength of the methodological design and theoretical framework, ethical considerations, description of thematic coding and theme generation, and the presentation of findings with supporting quotes.

## Results

Five datasets [30–34] and their associated publications [35–39] were included for appraisal and prompts were undertaken in September 2024. The duration of time from prompt to output ranged from 61 seconds to 123 seconds, with smaller datasets associated with shorter duration to produce an output. Three datasets [30,32,34] showed the time Copilot took to produce an output shortened each time it was prompted (Fig 1).

### Themes

Quotes and themes identified by human researchers in the published paper and Copilot are detailed in Supporting Information File 1: S1 Fig, S2 Table, and S3 Table. Overlap in themes identified by human researchers and Copilot were minimal, partly reflecting that in 2/5 studies the human researchers reported discursive thematic analyses referencing socio-cultural contexts while 5/5 Copilot outputs reported thematic analysis that did not reference socio-cultural contexts. One of the studies called their analysis 'content analysis' but reported themes and we could not identify any elements in the study that suggested thematic analysis had not been conducted [33] (Supporting Information File: S3 Table). Although Merkel et al. only made 15/54 interview transcripts available, the comparison of themes identified by human researchers and Copilot had considerable overlap [33].

In the three studies where both human researchers and Copilot reported thematic analysis, there was a reasonable synergy of themes reported by each. In Arora et al., for example, human researchers identified five themes, as did Copilot, with three from each being similar (i.e., essentially addressing the same theme) [35]. The five Arora et al. identified themes were: Implementation of peer education program, Connectivity between Health Programme-Rashtriya Kishor Swasthya Karyakram (RKSK) health workers and Peer Educators (PEs) and between PEs and adolescents during COVID-19 lockdown, Effect of COVID-19 on adolescent health services, Repurposing of RKSK health workers and PEs to

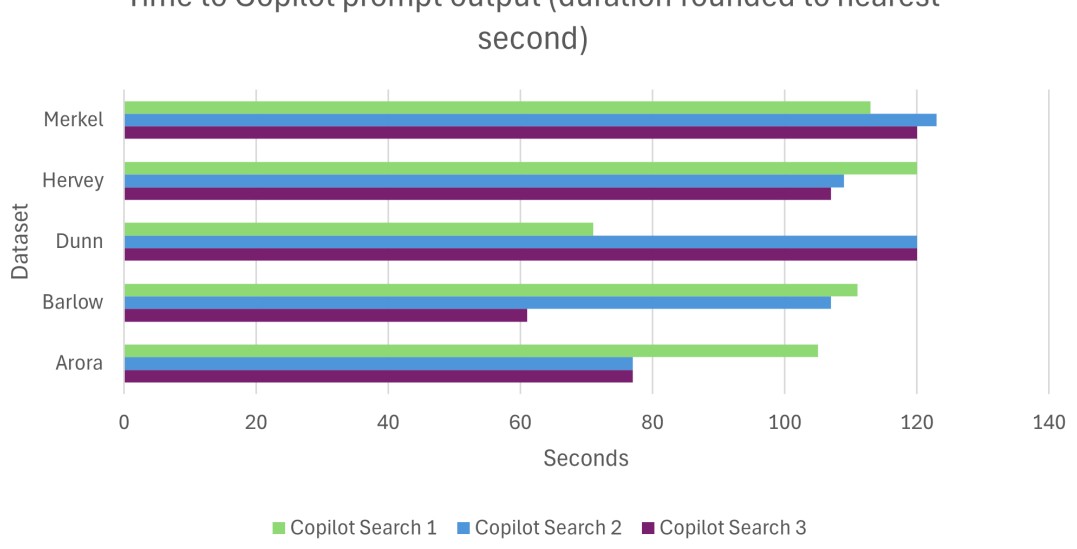

**Fig 1. Duration of GenAI to produce output following prompt.**

support COVID-19 response: During and post lockdown, and Adolescents' health and development issues during COVID-19. The five Copilot identified themes were: Implementation and challenges of the RKSK program, Impact of COVID-19 on program activities, Health issues faced by adolescents, Role of Peer Educators (PEs), and Differences in health issues between tribal and non-tribal adolescents (Supporting Information File 1: S3 Table). Neither the researchers (Arora et al.) nor Copilot reported the number of participants supporting each theme (called participant spread) [35]. Indeed, none of the Copilot outputs for the five studies reported the participant spread. We analyzed this manually and identified that participant spread was limited in Copilot outputs. Copilot outputs tended to draw most – if not all – of its reported themes and quotes from the first 2–3 pages of textual data, rather than from across the whole dataset (which in some cases were upwards of 150 pages).

### Quotes

Overall, while human researchers reported number of participants/documents and supported quotes with broad representation of quotes from many participants/documents, Copilot did not report number of participants/documents and provided fewer supporting quotes. We report the quote accuracy in terms of correct (verbatim), modified (quotes with some words changed but message unchanged) and fabricated (full or partial quote does not exist in dataset). Table 3 shows the different number of quotes provided by Copilot and human researchers. Table 4 indicates that human researchers provided substantially more correct quotes and less modified or fabricated quotes than Copilot. Quotes reported by human researchers were typically correct whereas over half of the quotes reported by Copilot were modified or fabricated (Table 5).

Copilot frequently modified (9.3%) or fabricated (44.5%) quotes, and failed to report participant numbers, often drawing themes from only the first few pages of data. Human researchers, while more accurate, also reported modified (11.2%) and potentially fabricated (20.5%) quotes. That Merkel et al.'s [38] full dataset was not provided meant we could not verify their reported quotes and these have been recorded as potentially fabricated though it is possible – even plausible – that their quotes are not fabricated; in which case the overall human researcher reporting of fabricated quotes is significantly lower than indicated here (i.e., the human researcher outputs were much more reliable than the

**Table 3. Themes identified by human researchers versus Copilot.**

| Study | Number of themes | | Total number of participants | | Quotes per theme | | Documents per theme | |
|---|---|---|---|---|---|---|---|---|
| | Human researcher analysis | Copilot analysis | Human researcher analysis | Copilot analysis | Human researcher analysis | Copilot analysis | Human researcher analysis | Copilot analysis |
| Arora, et al. 2022 [33] | 5 | 5 | 31 | Not stated | 5 | 2 | Not stated | Not stated |
| Barlow & Thow, 2021 [37] | 4 (discursive) | 5 | 47 | 47 | 2-8 | 2 | 2-8 | 1 |
| Dunn et al., 2021 [34] | 11 (5 sub-themes) | 8 | 33 | Not stated | 1-11 | 1 | Not stated | Not stated |
| Hervey, et al., 2021 [35] | 4 (discursive) | 8 | 48 | 15 | ≤ 1 | 2 | Not stated | Not stated |
| Merkel, et al., 2019 [36] | 11 | 7 | 54/54* | 5/15* | ~1 | 2 | Not stated | Not stated |

*The total number of interviews was 54, yet only 15 interview transcripts were published, and we used these 15 in the Copilot analysis, hence the different denominator in this comparison.

**Table 4. Human compared with Copilot accuracy of participant/document quotes.**

| Study | Absolute score (n) | | Relative score (%) | |
|---|---|---|---|---|
| | Published results | Copilot results | Published results | Copilot results |
| Quotes correct (Mean±SD) | 20.2±21.9 | 4.4±5.9 | 79.1±27.1% | 36.0±43.4% |
| Quotes modified (Mean±SD) | 2.2±3.5 | 0.6±0.9 | 6.2±11.2% | 6.5±9.3% |
| Quotes fabricated (Mean±SD) | 3.0±4.5 | 6.6±6.2 | 14.8±20.5% | 57.5±44.5% |

**Table 5. Examples of modified and fabricated quotes.**

| Original data | Modified/Fabricated Quote | Notes |
|---|---|---|
| "A study conducted in New Zealand in 2014 assessed the impact on consumer choice in relation to a selection of processed foods of four different types of food labelling and showed that traffic light food labelling was the best way of communicating the nutritional content of processed foods." (Barlow et al. dataset p56) [40] | Modified by Copilot: "The traffic light labelling system was the best way of communicating the nutritional content of processed foods." | This quote resembled the second half of the original quote. However, the original quote was considered a minor point by Barlow et al. and was not highlighted in their analysis. [39] |
| "We are working through WhatsApp only, everything is closed because of COVID-19. We are giving information to Saathiya/PE through WhatsApp only. All the information given to Saathiya/PE is related to the COVID-19 virus like what precautions they have to take, hand washing, wearing a mask when going out, and maintaining a distance of at least 6 feet when out." [30] | Modified by Arora et al.: "We are giving information to PEs through WhatsApp only. All information given is related to coronavirus, like what precautions they should take, frequent hand washing, wearing a mask when going out, cleaning used masks, and maintaining a distance of at least 6 feet." [35] | nil |
| Hervey et al.[32] | Fabricated by Copilot: "Workforce immigration is identified as an existential issue, especially for Scotland, where a significant proportion of healthcare workers are EU citizens." | This quote may have been a conglomeration of multiple statements in the original dataset into a single statement; however, we could not identify original statements from which it was potentially derived. |
| Arora. et al.[30] | Potentially fabricated by Arora: "We conduct outreach programmes in 5–6 schools every month but this year due to COVID-19, we haven't been able to do it." [35] | This quote may have been a conglomeration of multiple statements in the original dataset into a single statement; however, we could not identify original statements from which it was potentially derived. |

Copilot outputs). Furthermore, only one (20%) of the Copilot outputs across the five studies were free of quote fabrication, namely the Merkel dataset [38] with 14 quotes reflected accurately in the Copilot output (Supporting Information File: S2 Fig). Arora et al.'s study, which did have a full dataset available for secondary analysis, also reported quotes that could not be verified in their associated published dataset [35,38]. We emailed the authors [35,38] to seek clarification but received no response.

### Rigour

COREQ requires authors to note where they provide such information as *Supporting evidence: data used to support interpretation*, *Evidence of analytical process: Context described and taken account of in interpretation*, and *Additional techniques to enhance trustworthiness* [20,29]. Human researchers largely demonstrated these through their method and findings descriptions in their published articles. Both human researchers and Copilot provided data used to support interpretation, yet these are both subject to various levels of fabrication. Copilot did not provide evidence of analytical process or context considered during data interpretation.

Characteristics and reflexivity of the researchers were stated in the researcher publications and this informed the way data was collected and analysed, which remains unchanged in the Copilot outputs.

### Discussion

We set out to identify whether GenAI (Copilot) could be relied upon to provide accurate thematic analysis of existing previously published qualitative datasets. We identified five studies that met our inclusion criteria, and we carefully appraised both human researcher and Copilot analyses of the published datasets. At first glance, both human researcher and Copilot outputs seemed to have strong face validity. Yet, analyses of outputs by human researchers and Copilot unveiled

multiple errors and between-method discrepancies in results. First, we discuss the reliability of Copilot and human researcher outputs. Second, we compare our results with other studies whereby thematic analysis was undertaken using LLMs. Third, we discuss checklist use, transparency, rigour, and trustworthiness.

We took steps to minimize risk of bias informing the GenAI thematic analyses, including confirmatory bias, by selecting datasets for which researchers had already published their analysis (prior to 2022). Because we selected a closed GenAI system (Copilot), biases reflected in the GenAI outputs are limited to those reflected in the dataset or represented by the dataset, rather than the world wide web more broadly [41,42].

## Copilot and human researcher errors

Both Copilot and human researchers exhibited errors in their thematic analyses. While Copilot's issues were more pronounced in terms of quote fabrication and limited data utilization, the human research studies also had errors.

Additionally, the variability in analysis methods among human researchers led to minimal overlap with Copilot's themes. These findings underscore the need for rigorous validation and cross-checking in qualitative research, regardless of whether it is conducted by humans or AI.

The analytical capacity of GenAI is presently constrained to the identification of descriptive codes and themes. Contrary to common perceptions, GenAI systems do not possess autonomous cognitive functions. Rather, they operate by adhering to pre-programmed algorithms and instructions for data analysis. These systems lack the human capacity to interpret latent codes or uncover deeper meanings within the data. Consequently, their analysis remains confined to surface-level coding, which may account for the relatively frequent generation of fabricated quotes to substantiate the identified themes. Yet thematic analysis requires researchers to think and interpret. This disjunct between human researcher and GenAI capability is particularly evident in discursive thematic analysis; a finding consistent with other reported studies [26].

What is perhaps of even more concern is the human fabrication found in our analysis of the data sets. While GenAI does not know when it is making a mistake or fabricating results, humans likely do. This calls to question how much fabrication is occurring in qualitative research being conducted by humans. To reduce fabrication and increase transparency, the entire field of qualitative research could introduce requirements for publication of deidentified qualitative data sets upon which associated manuscripts are based. Examples of repositories that freely publish qualitative datasets include Open Science Framework and Mendeley. Tensions that such requirements would raise include those pertaining to ethical considerations concerning research methods (such as ethnographic film), as well as flows of power (such as datasets concerning minority groups, children, or people engaging in illegal activities; whereby stringent measures would be needed to protect participants (i.e., deidentify and anonymize data)).

## How these results compare with other studies

Previous studies have explored the use of GenAI tools in inductive and deductive thematic qualitative analysis [43,44]. These studies have focused on ChatGPT. Cook et al. and Mithas et al. have demonstrated that ChatGPT is useful in supporting stages of qualitative research, including transcription and translation, study planning, summarizing data sets and coding [43,44]. Further, there has been successful demonstration of using LLMs across the discrete stages of inductive and deductive thematic analysis in free-text response data [45] and in a variety of data contexts, such as semi-structured interviews with video gamers and university lecturers [46]. The benefits of using ChatGPT outlined by Cook et al., Mathas et al. and Morgan et al. align with the benefits of Copilot in this study, including the speed and accessibility of these tools. However, ChatGPT had comparable pitfalls to Copilot, including loss of nuance in analysis and the generation of misleading, inaccurate, and misaligned themes when compared to human-generated themes [26,43,44]. Together, these studies and ours suggest that while AI is not a replacement for human qualitative analysis, it may offer valuable support in some stages of qualitative research.

## Checklist use, transparency, rigour, and trustworthiness

In qualitative research, demonstrating research quality is imperative. Such demonstration forms the foundation of trust, upon which a project – and qualitative research more broadly – are deemed useful. Several techniques for demonstrating quality have been widely acknowledged in texts, checklists and guidelines.

In qualitative research, replicability is acknowledged as an impossibility because of the diversity of people and contexts. Heraclitus famously said: No man ever steps in the same river twice, for it's not the same river and he's not the same man [47]. Aligning to his wisdom, qualitative paradigms dictate that reproducing the same man [sic] or the same river is impossible. Given this, qualitative research instead strives for trustworthiness in terms of transparency about study design, data analysis, and steps taken to recognise and minimise researcher bias (often reported in terms of reflexivity or positionality) [48]. As mentioned in our introduction, the COREQ checklist is widely used to demonstrate rigour in studies drawing on interview and focus group data in medical and health sciences fields [29]. The more recent SRQR is widely used to report other qualitative research methods. In 2024, Braun and Clarke released their Big Q Qualitative Reporting Guidelines (BQQRG) to guide the analysis and reporting of qualitative research [10]. These tools guide researchers in how they conduct and report qualitative processes followed during data analysis. From our five included datasets, researchers variously described their analytical processes. However, the GenAI (Copilot) outputs were opaque. Next steps for quality appraisal of GenAI thematic analyses could include refinement prompting including details mirroring guiding frameworks such as COREQ or SRQR, or the BQQRG.

Such mirroring could serve to increase reliability and trustworthiness of analysis. However, it will not address the tension of quality that Braun and Clarke have warned against [10]. The COREQ and SRQR have been criticized as rigid, inhibiting transparency and fueling methodological incongruence. Psychologists – including Braun and Clarke – have raised concerns that such checklists promote 'little q' (qualitative research that is attempting to fit a positivist paradigm) rather than 'Big Q' (research that rejects objectivist assumptions, including those foundational to positivism) qualitative research. Acknowledging this tension, the 2024 BQQRG seeks to guide researchers to demonstrate quality while remaining true to qualitative philosophical paradigms. It is hard to see how researchers might successfully incorporate BQQRG use into future GenAI qualitative research for Big Q projects. Zhang and colleagues offer a prompt design framework for qualitative researchers to empower their analysis with GenAI and perhaps this offers a starting point [49].

Our study found that overlap in themes identified by human researchers and GenAI were typically minimal. In two of the five studies, human researchers reported discursive thematic analyses. The discursive approach is highly interpretive, and our prompt did not include a specific discursive component. A Chain-of-Thought prompt coupled with refinement prompting may have elicited discursive thematic analyses.

Discursive is also illustrative of Big Q qualitative approaches.

This article opened with reference to Mary Shelley's *Frankenstein*. Reflecting on our findings and the current international discourse concerning GenAI for qualitative research, we recommend caution. Pioneering a new way, as Shelley warns, evokes exploration of "unknown powers" [1]. Our exploration has been undertaken with human biases in mind. That is, human researchers want GenAI to make our research journeys easier, we want GenAI to produce reliable outputs. Our research has demonstrated that we must proceed cautiously, remaining alert to the potential for error and honing our human researcher capability for honing trustworthy GenAI use.

## Strengths and limitations

The strengths of this study are that five datasets were analyzed, representing a broad range of qualitative research, and the direct comparison between human researcher and GenAI (Copilot) thematic analyses. This approach allowed for a detailed examination of how each method (human researchers vs Copilot) identified and interpreted themes within the same datasets. By comparing the outputs, the study highlights the unique strengths and weaknesses of both human and AI analyses. This comparative framework provides valuable insights into the consistency, depth, and accuracy of

thematic identification, as well as the reliability of supporting quotes. It also helped to understand the potential biases and limitations inherent in AI-generated analyses, offering a comprehensive evaluation of GenAI's capabilities in qualitative research.

This study was limited to appraisal of a single LLM, which while widely used, was not custom designed for qualitative analysis. Our results may not be generalizable to other LLMs and may not apply to future iterations of Copilot. Regarding prompting, we elected not to include Few-Shot Learning prompting or Chain-of-Thought Reasoning prompting, nor refinement prompting in our methods. It is possible that these other forms of prompting and refinement prompting would have secured more accurate outputs (for example, to include quotes from a range of participants and to demonstrate this clearly in outputs).

Few studies met our inclusion criteria. We therefore opted to include all forms of thematic analysis studies, including discursive thematic analysis. Our prompt did not specifically mention discursive thematic analysis. We also opted to include Merkel et al. [31], despite the limitations borne of comparing analysis of 54 interviews (reported in their article) with analysis of 15 interviews (that authors provided through the UK Data Service). That the analyses reached the same themes supports the notion that data saturation can be reached within a small dataset.

We see this study as providing an important early step towards quality appraisal of GenAI thematic analyses. The results do not appear to be specific to health-related datasets. Next steps would be to develop refinement prompting into the methods to identify whether GenAI systems such as Copilot can be trained to reliably and accurately report different types of datasets according to thematic analysis. Similarly, Copilot did not provide evidence of analytical process or context considered during data interpretation despite the prompt element: *document the entire process transparently*. Next steps would be to develop refinement prompting regarding transparency of process.

## Conclusions

Face validity of Copilot thematic analysis looks promising, yet our analyses have shown that the current iteration of Copilot should not be relied upon for accurate or trustworthy thematic analysis. With effective prompting and close attention to outputs, Copilot accuracy may be improved upon to support thematic analysis.

## Supporting information

**S1 Table. Reported quotes by human researchers and genAI (Copilot).** *Merkel et al. made 15 interview transcripts available for analysis. We could not identify many of their reported quotes from the 15 interview transcripts and it is possible that the quotes come from the remaining transcripts that were not provided. This percentage should be considered with caution.
(DOCX)

**S2 Table. Thematic analysis: Human compared with GenAI Copilot thematic analysis.**
(DOCX)

**S1 Fig. Human compared with Copilot accuracy of participant/document quotes.**
(DOCX)

## Author contributions

**Conceptualization:** Tanisha Jowsey.

**Data curation:** Tanisha Jowsey, Shawna Campbell, Alexandra Davidson, Cher McGillivray, Isabella Maugeri, Megan Lee, Justin Keogh.

**Formal analysis:** Tanisha Jowsey, Peta Stapleton, Shawna Campbell, Alexandra Davidson, Cher McGillivray, Isabella Maugeri, Megan Lee, Justin Keogh.

**Methodology:** Tanisha Jowsey.

**Visualization:** Tanisha Jowsey.

**Writing – original draft:** Tanisha Jowsey, Peta Stapleton, Shawna Campbell, Alexandra Davidson, Cher McGillivray, Isabella Maugeri, Megan Lee, Justin Keogh.

**Writing – review & editing:** Tanisha Jowsey, Peta Stapleton, Shawna Campbell, Alexandra Davidson, Cher McGillivray, Isabella Maugeri, Megan Lee, Justin Keogh.

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
