## [Decision Letter · Decision Letter 0]

4 Jun 2025

PONE-D-25-06052Frankenstein, Thematic Analysis and Generative Artificial Intelligence: Quality Appraisal Methods and Considerations for Qualitative ResearchPLOS ONE

Dear Dr. Jowsey,

Thank you for submitting your manuscript to PLOS ONE. After careful consideration, we feel that it has merit but does not fully meet PLOS ONE’s publication criteria as it currently stands. Therefore, we invite you to submit a revised version of the manuscript that addresses the points raised during the review process.

We look forward to receiving your revised manuscript.

Kind regards,

Jiankun Gong

Academic Editor

PLOS ONE

Journal Requirements:

[The authors have no conflicts of interest to declare.].

3. In the online submission form, you indicated that [Some data are held in the UK Data Service repository, four of the five datasets have safeguarded restrictions.

Dataset 1 (Barlow et al.) ISSN 0277-9536, https://doi.org/10.1016/j.socscimed.2021.113761.

Dataset 2 (Hervey & Antova) UK Data Service SN: 854778, DOI: 10.5255/UKDA-SN-854778

Dataset 3 (Dunn et al) UK Data Service SN: 854245, DOI: 10.5255/UKDA-SN-854245

Dataset 4 (Holman & Walker) UK Data Service SN: 855082, DOI: 10.5255/UKDA-SN-855082

Dataset 5 (Arora) UK Data Service. SN: 855953, DOI: 10.5255/UKDA-SN-855953].

4. Please include captions for your Supporting Information files at the end of your manuscript, and update any in-text citations to match accordingly. Please see our Supporting Information guidelines for more information: http://journals.plos.org/plosone/s/supporting-information .

Reviewers' comments:

Reviewer's Responses to Questions

**Comments to the Author**

1. Is the manuscript technically sound, and do the data support the conclusions?

Reviewer #1: Partly

Reviewer #2: Partly

2. Has the statistical analysis been performed appropriately and rigorously? 

Reviewer #1: Yes

Reviewer #2: Yes

3. Have the authors made all data underlying the findings in their manuscript fully available?

Reviewer #1: Yes

Reviewer #2: Yes

4. Is the manuscript presented in an intelligible fashion and written in standard English?

Reviewer #1: Yes

Reviewer #2: Yes

5. Review Comments to the Author

Reviewer #1: This study is interesting; the aim (as you mentioned) is to determine the accuracy and efficiency of using generative artificial intelligence (GenAI) to undertake thematic analysis.

However, using Copilot for this task may be challenging, as it is difficult to change its parameters. It may be worth considering the application programming interface (API) of LLMs and adjusting to different versions of LLMs. For prompt engineering, there are zero-shot, few-shot, and chain-of-thought approaches.

Regarding the dataset, five datasets are quite small.

For result comparison, it would be better to illustrate more statistical results, such as accuracy, recall, precision, and F1 score.

In the discussion, you should compare the results with other thematic analysis using LLMs (of which there are quite a lot).

Reviewer #2: This paper explores the accuracy and credibility of Generative AI (GenAI), specifically Microsoft Copilot, in performing thematic analysis on existing qualitative datasets. The authors conduct a structured comparison using five real datasets and evaluate the differences between human and AI-generated themes based on accuracy, data support, and transparency. The topic is timely and relevant, and the comparative method offers a replicable way. However, the article needs revisions. I recommend that the suggestions be followed carefully to improve the scientific rigor of the work. It will be important to filter and prioritize the most relevant points.

Introduction

The introduction needs a clearer positioning of the research gap.

(a) The citations on prior GenAI-related studies lack critical thinking, and the gap remains vague. It would help to summarise existing attempts to apply GenAI in qualitative analysis and explain how this study makes a unique contribution in terms of method, objective, or evaluation dimension.

(b) The discussion on Big Q and Small q is overly long but not well connected to the design choices. For example, the authors do not explain why datasets involving discursive thematic analysis were included for AI comparison.

(c) The research aim is too broad. I suggest the authors clearly state their main goal at the end of the introduction.

Materials and Methods

The methods section describes the inclusion criteria and data sources in sufficient detail but still has several issues:

(a) The sample size is small, with only five studies selected. The authors should explain why the search was not extended to other journals or platforms.

(b) Copilot is chosen as the only GenAI tool, but there is no justification for excluding other tools like Claude or Gemini. A rationale is needed to explain why Copilot is suitable or representative for this task.

(c) The prompt used in Box 1 is central to the analysis, but the design process is unclear. The authors should explain how the prompt was developed and whether it was tested or reviewed before use.

Results

The results are clearly organized in tables, but the textual analysis lacks structure and detail.

(a) The paper would benefit from a clearer comparison framework across datasets. For example, the authors could compare theme alignment, data range, and expression style in a consistent way.

(b) Specific examples of themes and quotes generated by both Copilot and human researchers should be added to show differences more concretely.

(c) The criteria for verifying quotes should be explained. It is not clear how “fabricated” and “unverifiable” quotes were defined or distinguished.

Discussion

The discussion could reflect more critically on the design limitations of the study.

(a) The statement that Copilot fails to handle latent meaning is plausible, but currently based on assumption. It would be stronger if the authors gave concrete examples of failed interpretations or overlooked themes.

(b) The issue of quote accuracy in human studies is an important observation, but the evidence (i.e., lack of author response) is weak. The authors should clarify how they handled such cases to avoid confusion or unfair judgment.

Rigour and Trustworthiness

On page 7, the authors briefly mention the use of COREQ to assess analysis transparency, data support, and participant representation. However, there is no clear mapping of COREQ items to the evaluation results. I recommend providing a supplementary table showing which of the 32 COREQ items were met in human and AI outputs. A simple ✓ / ✗ or short comment would improve clarity.

Conclusion

The conclusion is too general and lacks clear boundary conditions. The authors should clarify whether their findings apply only to health-related datasets and whether the results can be generalized to other LLMs.

6. PLOS authors have the option to publish the peer review history of their article (what does this mean? ). If published, this will include your full peer review and any attached files.

**Do you want your identity to be public for this peer review?** For information about this choice, including consent withdrawal, please see our Privacy Policy .

Reviewer #1: No

Reviewer #2: No

---

## [Author Response · Author response to Decision Letter 1]

11 Jul 2025

Thank you to the reviewers. Below are the reviewer comments and our responses. The reviewers may find it easier to read these responses in the response letter we have uploaded.

COMMENT RESPONSE

Reviewer 1:

Reviewer #1: This study is interesting; the aim (as you mentioned) is to determine the accuracy and efficiency of using generative artificial intelligence (GenAI) to undertake thematic analysis.

However, using Copilot for this task may be challenging, as it is difficult to change its parameters. It may be worth considering the application programming interface (API) of LLMs and adjusting to different versions of LLMs. For prompt engineering, there are zero-shot, few-shot, and chain-of-thought approaches.

Thank you for these comments. The reviewer kindly suggests using the API to adjust different versions of LLMs and mention the type of prompt engineering we used. We only explored Copilot and no other LLMs because of the restrictions placed on us by our university and the UK Data Service. We have now explained this more clearly in the manuscript. We have now stated our promoting approach as zero-shot in the manuscript, and listed the limitations associated therewith in our limitations section, thank you for bringing that to our attention.

Regarding the dataset, five datasets are quite small.

We suggest that five datasets is still more than previous studies in this area. Morgan’s study, which we cite as precedence, analysed two datasets. We sought to identify more than five studies for inclusion in the research but only five studies met our criteria.

For result comparison, it would be better to illustrate more statistical results, such as accuracy, recall, precision, and F1 score.

We appreciate that recall, precision and F1 score (which is a metric that includes recall and precision) play important roles in quantitative studies involving classification of true and false positives and negatives, particularly in machine learning studies. However, as we analysed qualitative rather than quantitative data involving multiple levels of classification of true and false positives and negatives, it was apparent that metrics including recall, precision and F1 score would not be appropriate for the data we collected.

We do present accuracy measures including how often Co-pilot got Number of themes, Total number of participants, Quotes per theme and Documents per theme correct compared to humans.

In the discussion, you should compare the results with other thematic analysis using LLMs (of which there are quite a lot).

This has been done as suggested.

Reviewer 2:

This paper explores the accuracy and credibility of Generative AI (GenAI), specifically Microsoft Copilot, in performing thematic analysis on existing qualitative datasets. The authors conduct a structured comparison using five real datasets and evaluate the differences between human and AI-generated themes based on accuracy, data support, and transparency. The topic is timely and relevant, and the comparative method offers a replicable way. However, the article needs revisions. I recommend that the suggestions be followed carefully to improve the scientific rigor of the work. It will be important to filter and prioritize the most relevant points.

Introduction

The introduction needs a clearer positioning of the research gap.

(a) The citations on prior GenAI-related studies lack critical thinking, and the gap remains vague. It would help to summarise existing attempts to apply GenAI in qualitative analysis and explain how this study makes a unique contribution in terms of method, objective, or evaluation dimension.

(b) The discussion on Big Q and Small q is overly long but not well connected to the design choices. For example, the authors do not explain why datasets involving discursive thematic analysis were included for AI comparison.

(c) The research aim is too broad. I suggest the authors clearly state their main goal at the end of the introduction.

We thank the reviewer for their comments and have made the following provisions as requested:

a) We have now made clear the unique contribution of our study makes in relation to existing literature;

b) We have improved integration of the discussion on Big Q and small q throughout methods and have made clear in methods that discursive thematic analyses were included;

c) the main goal and research question are now described at the end of the introduction.

Materials and Methods

The methods section describes the inclusion criteria and data sources in sufficient detail but still has several issues:

(a) The sample size is small, with only five studies selected. The authors should explain why the search was not extended to other journals or platforms.

We suggest that five datasets is a lot. Morgan’s study, which we cite as precedence, analysed two datasets. We sought to identify more than five eligible studies for inclusion in the research but only five studies met our criteria. The amount of quality appraisal work associated with our study was substantial, so we were actually grateful that no further studies met our criteria.

(b) Copilot is chosen as the only GenAI tool, but there is no justification for excluding other tools like Claude or Gemini. A rationale is needed to explain why Copilot is suitable or representative for this task. Thank you for pointing this out. We have now clarified that we didn’t have a whole lot of choice. Our institution and the UK Data Service would only approve Copilot for this study. Co-pilot is our institutionally supported closed GenAI tool. Other tools are not “closed” systems and thus breach the data privacy statements of the UK database. Please see below GenAI policy from Bond University https://bond.edu.au/current-students/study-information/integrity-at-bond/academic-integrity/academic-integrity-and-artificial-intelligence

Text now reads:

The GenAI tool we selected was Copilot on the grounds that it was a general-purpose GenAI, approved for use by both our institution and the UK Data Service, and was a closed GenAI system (i.e. access to Copilot is limited and password protected, so that the data are protected and are not used to train foundation GenAI models). The UK Data Service permitted us to include in our study data published through their repository but only for analysis through approved closed GenAI systems (i.e. Copilot) and not for open GenAI systems like ChatGPT, Perplexity, Claude or Gemini.

(c) The prompt used in Box 1 is central to the analysis, but the design process is unclear. The authors should explain how the prompt was developed and whether it was tested or reviewed before use. Good point. We have added a sentence regarding the design process of developing the prompt.

Text now reads:

Prior to running the prompt through Copilot, the team developed, tested, refined, reviewed and piloted the prompt wording.

Results

The results are clearly organized in tables, but the textual analysis lacks structure and detail.

(a) The paper would benefit from a clearer comparison framework across datasets. For example, the authors could compare theme alignment, data range, and expression style in a consistent way.

Theme alignment and data range are detailed in Supplementary File 1. No change made. However, we thank the reviewer for pointing us to look again at this because we realised we had not included quotes in the manuscript or Supplementary File. We have now included quote examples in the manuscript Table 4.

(b) Specific examples of themes and quotes generated by both Copilot and human researchers should be added to show differences more concretely. We provide all themes in Supplementary File 1, and have now included an example of these in the manuscript too. We now include examples of quotes in the manuscript.

(c) The criteria for verifying quotes should be explained. It is not clear how “fabricated” and “unverifiable” quotes were defined or distinguished. Thank you, we have now clarified that the two terms we are using are modified and fabricated; modified had words changed or missing and fabricated quotes were not in the original datasets either in part or in full. Examples now provided in Table 4.

Discussion

The discussion could reflect more critically on the design limitations of the study.

(a) The statement that Copilot fails to handle latent meaning is plausible, but currently based on assumption. It would be stronger if the authors gave concrete examples of failed interpretations or overlooked themes.

(b) The issue of quote accuracy in human studies is an important observation, but the evidence (i.e., lack of author response) is weak. The authors should clarify how they handled such cases to avoid confusion or unfair judgment.

We thank the reviewer for their comments and have made the following changes:

a) The statement regarding how Copilot failed to handle latent meaning has been revised; all themes that were identified by researchers but not by Copilot are listed in Supplementary File 1.

b) That you for drawing out attention to this. The section regarding quote accuracy in human studies has been substantially revised providing greater evidence and clarity to the reader on how these cases were handled and recommending caution for interpretation.

Rigour and Trustworthiness

On page 7, the authors briefly mention the use of COREQ to assess analysis transparency, data support, and participant representation. However, there is no clear mapping of COREQ items to the evaluation results. I recommend providing a supplementary table showing which of the 32 COREQ items were met in human and AI outputs. A simple ✓ / ✗ or short comment would improve clarity.

Conclusion

After careful consideration, we have decided not to provide the full 32 item assessment against researcher and Copilot outputs on the grounds that COREQ is a reporting checklist for writing specific sections of qualitative manuscripts and not particularly for assessing the quality of the data or analysis itself.

Many qualitative manuscripts do not report COREQ. We acknowledge that AI would perform poorly on this task as we did not prompt it to follow COREQ guidelines.

The conclusion is too general and lacks clear boundary conditions. The authors should clarify whether their findings apply only to health-related datasets and whether the results can be generalized to other LLMs. Conclusion has been tightened. We also now state that the results do not appear to be specific to health-related datasets.

Thank you for your comments. We trust our revisions meet with your approval.

---

## [Decision Letter · Decision Letter 1]

29 Jul 2025

Frankenstein, Thematic Analysis and Generative Artificial Intelligence: Quality Appraisal Methods and Considerations for Qualitative Research

PONE-D-25-06052R1

Dear Dr. Jowsey,

We’re pleased to inform you that your manuscript has been judged scientifically suitable for publication and will be formally accepted for publication once it meets all outstanding technical requirements.

Kind regards,

Jiankun Gong

Academic Editor

PLOS ONE

Additional Editor Comments (optional):

Reviewers' comments:

Reviewer's Responses to Questions

**Comments to the Author**

1. If the authors have adequately addressed your comments raised in a previous round of review and you feel that this manuscript is now acceptable for publication, you may indicate that here to bypass the “Comments to the Author” section, enter your conflict of interest statement in the “Confidential to Editor” section, and submit your "Accept" recommendation.

Reviewer #2: All comments have been addressed

2. Is the manuscript technically sound, and do the data support the conclusions?

Reviewer #2: Yes

3. Has the statistical analysis been performed appropriately and rigorously? 

Reviewer #2: Yes

4. Have the authors made all data underlying the findings in their manuscript fully available?

Reviewer #2: Yes

5. Is the manuscript presented in an intelligible fashion and written in standard English?

Reviewer #2: Yes

6. Review Comments to the Author

Reviewer #2: Dear Authors, Thanks very much for your revised submission. I’ve read the updated version of the manuscript titled "Frankenstein, Thematic Analysis and Generative Artificial Intelligence: Quality Appraisal Methods and Considerations for Qualitative Research."

You've clearly taken the feedback seriously. The revised paper is much more focused and easier to follow. The methods section is now clearer, and I can see a stronger link between your analysis and the questions you're trying to answer. The explanation of how GenAI was tested, and the comparison with human analysis, is now much more solid and convincing.Overall, the paper makes an original contribution, and I hope readers will find it both relevant and thought-provoking.

7. PLOS authors have the option to publish the peer review history of their article (what does this mean? ). If published, this will include your full peer review and any attached files.

**Do you want your identity to be public for this peer review?** For information about this choice, including consent withdrawal, please see our Privacy Policy .

Reviewer #2: **Yes: ** Shuangyan Du

---

## [Editor Report · Acceptance letter]

PONE-D-25-06052R1

PLOS ONE

Dear Dr. Jowsey,

I'm pleased to inform you that your manuscript has been deemed suitable for publication in PLOS ONE. Congratulations! Your manuscript is now being handed over to our production team.

Kind regards,

on behalf of

Dr. Jiankun Gong

Academic Editor

PLOS ONE